# Acute, Sublethal, and Developmental Toxicity of Kratom (*Mitragyna speciosa* Korth.) Leaf Preparations on *Caenorhabditis elegans* as an Invertebrate Model for Human Exposure

**DOI:** 10.3390/ijerph19106294

**Published:** 2022-05-22

**Authors:** Samantha Hughes, David van de Klashorst, Charles A. Veltri, Oliver Grundmann

**Affiliations:** 1A-LIFE Amsterdam Institute for Life and Environment, Section Environmental Health and Toxicology, Vrije Universiteit Amsterdam, 1081 HV Amsterdam, The Netherlands; s.hughes@vu.nl; 2HAN BioCentre, HAN University of Applied Sciences, 6525 EM Nijmegen, The Netherlands; d.klashorst@hotmail.com; 3Department of Pharmaceutical Sciences, College of Pharmacy, Midwestern University, Glendale, AZ 85308, USA; cveltri@midwestern.edu; 4Department of Medicinal Chemistry, College of Pharmacy, University of Florida, Gainesville, FL 32610, USA

**Keywords:** opioid, toxicity, pharyngeal pumping, body bending, reproduction

## Abstract

Kratom (*Mitragyna speciosa* Korth.) is a tree native to Southeast Asia with stimulant and opioid-like effects which has seen increased use in Europe and North America in recent years. Its safety and pharmacological effects remain under investigation, especially in regard to developmental and generational toxicity. In the current study, we investigated commercial kratom preparations using the nematode *Caenorhabditis elegans* as a translational model for toxicity and pharmacological effects. The pure alkaloids mitragynine and 7-hydroxymitragynine as well as aqueous, ethanolic, and methanolic extracts of three commercial kratom products were evaluated using a battery of developmental, genotoxic, and opioid-related experiments. As determined previously, the mitragynine and 7-hydroxymitragynine content in kratom samples was higher in the alcoholic extracts than the aqueous extracts. Above the human consumption range equivalent of 15–70 µg/mL, kratom dose-dependently reduced brood size and health of parent worms and their progeny. 7-hydroxymitragynine, but not mitragynine, presented with toxic and developmental effects at very high concentrations, while the positive control, morphine, displayed toxic effects at 0.5 mM. Kratom and its alkaloids did not affect pumping rate or interpump interval in the same way as morphine, suggesting that kratom is unlikely to act primarily via the opioid-signalling pathway. Only at very high doses did kratom cause developmental and genotoxic effects in nematodes, indicating its relative safety.

## 1. Introduction

The tree commonly referred to as kratom (*Mitragyna speciosa* Korth., Rubiaceae) is native to Southeast Asia, primarily growing in Thailand, Malaysia, and Indonesia, and has been exported worldwide [1]. The traditional use of kratom dates back centuries for a range of conditions, including gastrointestinal disorders, fever, and acute and chronic pain, as well as a mild stimulant used by day laborers [2] Due to its increased use outside of Southeast Asia, and especially in North America and Europe, over the past decade kratom has been studied for its pharmacological effects [3]. The fresh leaves are primarily used in its native environment to be chewed or made into a tea while dried leaf material is exported to be used in various preparations for oral consumption [4]. The primary focus of research on kratom has been on its indole and oxindole alkaloids with mitragynine viewed as the most abundant alkaloid [5]. The pharmacological effects of mitragynine appear to be mediated through a number of pathways, including interaction with opioid, adrenergic, dopaminergic, and serotonergic receptors [6,7,8]. Mitragynine acts as a partial, biased agonist at μ opioid receptors and a competitive antagonist at both κ- and δ-opioid receptors [6,9,10]. The close structural analogue and metabolite, 7-hydroxymitragynine, is a more potent partial agonist at the μ opioid receptors but shows similar competitive antagonism at the other opioid receptors [9,11,12].

Kratom is used by consumers as self-treatment for a wide range of health conditions, including psychiatric disorders (depression, anxiety, attention deficit hyperactivity disorder, post-traumatic stress disorder), acute and chronic pain (lower-back pain, arthritis, neuropathic pain), and mitigation of withdrawal symptoms from a prescription or illicit drug dependence [3,13,14,15]. However, the variability in commercial kratom products available to consumers complicates determination of the active principles responsible for the chosen self-treatment indication. The usual dose range consumed by the majority of users is between 1 and 5 g taken up to 3 times/day [13]. It is unclear if acute or chronic consumption of kratom at these doses leads to toxic effects in humans. Several case reports do associate kratom exposure to adverse effects, primarily affecting the cardiovascular, hepatic, and central nervous system [16,17,18]. Acute and subchronic liver injury have been reported, although without further information on the composition of the kratom products [17]. Prolonged QTc intervals have also been observed in electrocardiograms of regular kratom users, although they do not reach pathological levels [19]. In addition, effects of kratom on the developing foetus remains unclear, with one study indicating developmental toxicity in zebrafish embryos of a fresh leaf water decoction (≥500 μg/mL), mitragynine, and speciociliatine (both ≥50 μg/mL) at high concentrations [20]. Consequently, a correlation between kratom toxicity, the contribution of respective alkaloids, and human dose ranges requires further investigation.

The nematode worm *Caenorhabditis elegans* is an increasingly popular test organism for toxicity screening [21,22,23,24,25] due to its strong predictive power [26,27]. The 1 mm-long nematode has a short and well-characterised life cycle of 3 days at 20 °C, and the self-fertilising hermaphrodite is capable of producing 250–300 genetically identical offspring [28]. Despite its simplicity, there is great conservation at the genetic, cellular, tissue, and organ levels between *C. elegans* and mammals [29,30,31]. Additionally, the worm has an extremely well-characterised nervous system [32,33] with the neurons, neuromuscular junctions, and synapses being the same as those found in humans, further highlighting the translational relevance of *C. elegans* to humans [34].

Strikingly, *C. elegans* responds to a variety of substances that are commonly abused by humans, including cocaine, methamphetamine, morphine, and alcohol, in a fashion similar to rodents [35,36,37]. Further, an opioid signalling system has been identified in *C. elegans* that has been shown to respond to opioid receptor agonists [38,39,40]. Together, this evidence demonstrates that *C. elegans* is a useful model for exploring the effects of kratom toxicity. Due to the conservation at the cellular level with humans [31,41] as well as having conserved biological processes [42,43], the physiological responses of *C. elegans* following exposure to compounds such as kratom are a good reflection of the overall response expected in humans, and provide key insights into the molecular mechanisms of toxicity [44].

In the current study, we investigated the effects of kratom on the nematode *C. elegans*, evaluating the acute and developmental toxicity of commercial kratom preparations. We chose to explore methanolic and ethanolic extracts, as these are conventional extraction solvents used to prepare samples for clinical research, as well as water extracts as these are more in line with human ingestion. In addition, we explored the effect of the pure alkaloids mitragynine and 7-hydroxymitragynine. Given the difference in kratom preparations, we sought to contrast an aqueous decoction with commonly studied ethanolic and methanolic extracts with regard to both alkaloid content and observed toxicity. To our knowledge, this is the first study to investigate toxicity of kratom in *C. elegans* as a feasible model to reflect human exposure.

## 2. Materials and Methods

### 2.1. Materials

Mitragynine and 7-hydroxymitragynine stock solutions were obtained from Sigma-Aldrich (St. Louis, MO, USA). ACS-grade 200-proof ethanol and methanol and LCMS-grade acetonitrile, water, and formic acid were purchased from Fisher Scientific (Waltham, MA, USA). The kratom products were purchased from a tobacco and smoke shop in Phoenix, AZ, in July and August of 2019. The samples were stored in the dark, at 4 °C, in their original packaging, until extraction. For the nematode experiments, serotonin hydrochloride was purchased from Sigma Aldrich (#H9523), as was DMSO (#D1435). Morphine was purchased at 10 mg/mL morphine hydrochloride (Kalceks) in a physiological buffer solution. This equated to a 23.6 mM solution.

### 2.2. Extraction of Kratom Powder

Microwave-assisted extraction (MAE) was used as a way to exhaustively extract semi-volatile compounds due to the increased efficiency and throughput, coupled with decreased extraction time associated with this method. All three solvents were extracted in the same manner to limit additional influence of extraction efficiency inherent to the three solvents. Three commercial kratom powder products (Earth Kratom Organic Red Maeng Da, White Borneo, and Bali distributed by Fire Wholesale) were used to prepare the extracts. The samples (1 g) were placed in individual teabags and extracted with 12 mL of 100% methanol, ethanol, or milliQ water using microwave-assisted extraction with an Ethos EX labstation (Milestone Srl, Sorisole, Bergamo, Italy). The extraction conditions were a 15 min ramp to boiling temperature and maintaining boiling temperature for 30 min, followed by a 15 min cool-down, for a total run time of 1 h. The extracts were gravity-filtered, dried to completion by centrifugal evaporation, and stored at 4 °C until chemical analysis.

### 2.3. Analysis of Kratom Extracts

For evaluation of mitragynine and 7-hydroxymitragynine concentrations, extracts were diluted to 0.1 mg/mL and 25 mg/mL, respectively. Extracts were placed in the autosampler rack of an Agilent 1260 HPLC system (Agilent Technologies, Santa Clara, CA, USA). For each HPLC separation, 1 µL was injected and chromatographed on an Agilent Poroshell 120 EC-C18 column (4.6 × 100 mm, 2.7 µm, Agilent Technologies, Santa Clara, CA, USA), equipped with the appropriate guard column, and separated at a flow rate of 0.5 mL/min. The mobile phase consisted of eluent A (LCMS-grade water with 0.1% formic acid) and eluent B (LCMS-grade acetonitrile with 0.1% formic acid). An isocratic elution of 25% B was started for 1 min, followed by a linear gradient applied from 25% to 75% B over 11 min, and finally an isocratic wash of 75% B for 5 min was applied before a re-equilibration of 25% B for 3 min. Typically, a back pressure of <95 bar was observed at 25% acetonitrile–water.

Compounds were detected using an Accurate-Mass 6530 quadrupole time-of-flight mass analyser (Q-TOF, Agilent Technologies, Santa Clara, CA, USA). The Q-TOF mass analyser provided identification of compounds using accurate masses of full spectra in targeted MS/MS mode. The transition used to quantify mitragynine was the precursor ion of 399.2 *m*/*z* fragmented into the product ion of 174.1 *m*/*z* (collision-induced dissociation = 30 V). The transition used to quantify 7-hydroxymitragynine was the precursor ion of 415.2 *m*/*z* fragmented into the product ion 190.1 *m*/*z* (collision-induced dissociation = 30V). The retention times for mitragynine and 7-hydroxymitragynine were 5.31 and 2.45 min, respectively.

Quantification of mitragynine and 7-hydroxymitragynine was performed using a series of standard curves prepared fresh on each day of analysis at incremental concentrations ranging from 10 ng/mL to 100 µg/mL using a quadratic equation with weighting 1/x. The regression coefficient for both standard curves was ≥0.99 with repeatability (intraday variability) ranging from 3.3 to 5.3% for 7-hydroxymitragynine and from 2.1 to 15.7% for mitragynine while intermediate precision (interday variability) ranged from 6.9 to 14.9% for 7-hydroxymitragynine and from 5.3 to 16.6% for mitragynine (Table 1). All samples were analysed using MassHunter Software (Agilent Technologies, Santa Clara, CA, USA).

### 2.4. C. elegans Strain and Culture Conditions

The *C. elegans* strain used in this study was *N2* (wild-type var. Bristol) provided by the *Caenorhabditis* Genetics Centre (CGC). Worms were maintained at 20 °C on Nematode Growth Medium (NGM) seeded with a lawn of *E. coli OP50*, also provided by the CGC, according to standard protocols [45].

Bleaching was used to age-synchronise nematodes before each experiment to generate a synchronous population [46]. In brief, gravid worms were subjected to alkaline hypochlorite solution (2% *w*/*v* sodium hypochlorite, 400 mM sodium hydroxide) to release the fertilised embryos. Hypochlorite solution was removed by washing with M9 buffer (22 mM potassium dihydrogen phosphate, 42 mM sodium hydrogen phosphate, 86 mM sodium chloride, and 1 mM magnesium sulphate). Eggs were left to hatch overnight at room temperature (19–21 °C) in M9 buffer in the absence of a food source, resulting in a population of synchronised L1 larvae. Once plated on *OP50*-seeded NGM, the worms developed as normal.

### 2.5. Preparation of Kratom Extracts

Prior to biological analysis, all kratom extracts were stored at −20 °C until required. The solid extracts were diluted in 100% DMSO to a concentration of 50 mg/mL. The samples were left overnight at room temperature (19–21 °C) on a rocking platform to fully dissolve.

For the dose–response curve in S-complete [47], the relevant amount of stock solution was added to S-complete before being added to the 96-well plates. For the solid NGM, stock solution was added directly to the cooled molten NGM prior to pouring the plates, to a final concentration of 300 µg/mL. The plates were left overnight, protected from light, before being seeded with a lawn of *E. coli OP50*. The following day plates were used. Plates were always made within 2 days of being used.

### 2.6. Analysis of Developmental and Reproductive Toxicity in C. elegans

To explore the concentration at which the different varieties and extracts of kratom had a detrimental effect on development and reproduction, a dose–response assay was performed in S-complete liquid culture, containing kratom from 15 to 400 µg/mL. The dose range was based on an *in vitro* study using kratom and its alkaloids [48]. A single L4-staged wild-type animal in M9 buffer was added to each well of a 96-well plate containing S-complete, bacteria, and kratom, where each row was a different concentration with up to 12 replicates per condition. Each plate included a negative control and a solvent control. The plates were incubated at 20 °C with gentle shaking at 150 rpm. After 48 h, each well was assessed for acute toxicity of the parent as well as a reproductive effect.

As this is a method of screening for toxicity, the observation of each well was given a broad classification. For acute toxicity, the parent worms were classified into three groups: ‘Healthy’, where the worm displayed normal development and head-to-tail body bending; ‘slightly abnormal’, indicating that there was a small impact on mobility; and lastly, ‘Dead or not moving’, where the worm was observed as being dead by having a rod-like appearance, or close to death where there was very infrequent motion and usually only of the head. An estimate of the brood size was also recorded as 1 of 4 groups: no viable progeny; up to 10 viable offspring; 11–75 viable progeny; and more than 75 viable progeny (i.e., wild type).

### 2.7. Thrashing Analysis

To assess thrashing (head-to-tail body bends), L4-staged *C. elegans* were added to kratom (all varieties and extracts at 300 µg/mL)-supplemented NGM plates and incubated at 20 °C for 48 h. A DMSO control and negative control were also included. After 2 days, individual worms were placed into M9 buffer (~40 µL) on an unseeded NGM plate at room temperature. Video recordings (90 s) of the nematodes were acquired using a Leica S8aP0 binocular microscope with a Leica DMC2900 camera using the LAS v4.12 acquisition software. The videos were analysed using ImageJ v1.53, and the number of body bends was assessed using the wrMTrck plugin (build 110622) [49,50]. Videos of at least 20 worms in each condition were analysed with the average body bends of each group calculated and normalised to the control conditions so that the number of body bends per minute for the controls was 100%. Graphs were prepared and analysed in GraphPad Prism v9.2.0.

### 2.8. Pharyngeal Pumping

Synchronised worms were grown on NGM plates seeded with *E. coli OP50*. When nematodes reached the L4 stage, they were transferred to seeded NGM supplemented with 300 µg/mL White Borneo kratom extract (methanol, ethanol, or water extracts). A positive control of 0.5 mM morphine was used based on the literature [40], and a negative solvent control of DMSO was included. Plates were incubated at 20 °C for 48 h, after which worms were either directly prepared for analysis of pumping rate (well-fed) or transferred to non-seeded NGM supplemented with kratom, morphine, or DMSO for 1 h at 20 °C and then prepared for pumping analysis (starved).

Pumping rates were measured as previously described [51,52]. In brief, worms were washed off the plates with M9 buffer and washed twice with M9 buffer by centrifugation (2500 rpm, 90 s). After washing, worms were incubated for 10 min in 10 mM serotonin in M9 to stimulate pumping. Electropharyngeogram (EPG) recordings of 2 min were taken using the NemaMetrix ScreenChipTM system using an SC30 chip and the NemAquire v2.1 software. NemAnalysis-0.2 software was used to analyse the EPGs.

### 2.9. Data and Statistical Analysis

The data and statistical analysis comply with the recommendations on experimental design and analysis in pharmacology. For the established analytical method, at least three separate standard curves were prepared for each run on at least three separate days to determine the intraday and interday coefficient of variation (CV) as the intraday or interday standard deviation divided by the mean expressed as a percentage at each concentration of mitragynine and 7-hydroxymitragynine. For pharyngeal pumping, averages were calculated for pump duration, interpump interval and pumps per minute with a one-way ANOVA used for statistical analysis. Graphs of pumps per minute were prepared in GraphPad Prism v9.2.0. Representative images of the pumps were extracted from the NemAnalysis-0.2 software. For the well-fed conditions, at least 10 worms were analysed across 2 independent experiments, and the data were combined. For the starved worms, a minimum of 5 worms were analysed for each condition in one independent experiment. Statistical significance was set to *p* < 0.05.

## 3. Results

### 3.1. Mitragynine and 7-Hydroxymitragynine Content in Kratom Samples Depended on Extraction Solvent

Using the established LC-QTOF method, methanolic and ethanolic extracts prepared from White Borneo and Bali kratom powders presented with the highest concentration of mitragynine (Table 2). The highest concentration of 7-hydroxymitragynine was found in the methanolic extract for Red Maeng Da (0.052%), whereas Bali and White Borneo presented much lower concentrations of 7-hydroxymitragynine in the methanolic extract (0.011% and 0.012%, respectively). The water extracts were low in both mitragynine (0.1–0.8%) and 7-hydroxymitragynine (0–0.014%) (Table 2). The representative chromatograms using Red Maeng Da extractions (Figure 1) further exemplify the increased complexity of the alcoholic extractions compared to the more traditional water extraction. 

### 3.2. Kratom Dose-Dependently Reduced Brood Size and Health of Parents and Progeny

To explore toxicity of kratom exposure, a dose–response curve was prepared in liquid culture, where each kratom extract was tested at concentrations from 15 to 400 µg/mL. To assess acute toxicity, the parent worms were observed after 48 h of exposure to kratom extracts (Figure 2). Water extracts of White Borneo kratom were the least toxic to the parent, while exposure to water extracts of Bali and Red Maeng Da kratom caused some impairment at concentrations above 45 µg/mL. Both ethanolic and methanolic extracts of Bali and Red Maeng Da led to impaired movement in concentrations above 25 µg/mL. The methanolic and ethanolic extracts of White Borneo kratom resulted in a more striking effect on parent health, with exposure to these extracts causing death in parent worms above a concentration of 200 µg/mL.

All parent worms were able to produce progeny, and the number of offspring was assessed in each exposure condition (Figure 3 and Appendix A). There was a prominent decrease in the number of viable progeny in worms that were exposed to White Borneo kratom extracts, with the effects observed from 45 µg/mL (Figure 3A). In contrast, a small decrease in brood size was observed with increasing concentrations of Red Maeng Da and Bali extracts, and these effects were observed at the higher concentration of exposures, from 350 µg/mL (Figure 3B,C).

To compare the effect of kratom to a known opioid compound, we used morphine at a concentration of 0.5 mM as a positive control, based on literature values [40]. Strikingly, morphine did not display acute toxicity to the parent, nor affect the development of the progeny or reproduction (Appendix A).

### 3.3. 7-Hydroxymitragynine Did Present with Toxic and Developmental Effects

It was clear that the White Borneo kratom had more of an impact on the development and reproductive toxicity of the nematodes compared to the other kratom varieties tested. It was therefore of interest to explore if these effects were due to the alkaloids present in the White Borneo samples. Comparison of mitragynine and 7-hydroxymitragynine in brood size over a concentration range from 0.1 to 5 µg/mL indicated a significant toxic effect of 7-hydroxymitragynine at the highest concentration tested, whereas mitragynine did not affect brood size (Figure 4A). The solvent control, methanol, did not impact brood size. Exposure to 0.5 mM morphine resulted in a slight reduction in brood size compared to controls (Figure 4B). 7-hydroxymitragynine exposure impaired the health of both the progeny (Figure 4C) and the parents (Figure 4D) at high concentrations, even leading to death. Exposure to mitragynine did impact the parents and progeny, although not to the same extent as 7-hydroxymitragynine (Figure 4C,D). It should be noted that although parent deaths were observed at the lowest concentration of 0.1 µg/mL mitragynine and the highest concentration of 7-hydroxymitragynine (Figure 4D), this was in fact due to the death of a single worm.

### 3.4. Morphine but Not Kratom Displayed Toxic Effects

The dose–response assay indicated that the kratom caused a slight impairment of mobility, which is suggestive of a toxic effect. We chose two assays to explore this: a mobility assay [54,55,56] and a pharyngeal pumping assay [57,58,59].

Mobility is a powerful indicator of health and can be used to assess the impact of drugs [56]. To this end, worms were exposed to controls (a DMSO solvent control and 0.5 mM morphine) or 300 µg/mL kratom for 48 h. The number of body bends was assessed and normalised to the controls, such that the controls were set to 100%. Morphine significantly (*p* < 0.001) lowered body bends (thrashing) compared to control animals (Figure 5). Neither the Red Maeng Da or Bali kratom preparation altered thrashing significantly in any solvent preparation, whereas White Borneo water and methanol extracts significantly lowered body bends (Figure 5 and Appendix A).

As exposure to White Borneo caused the most striking effects in *C. elegans*, we chose to explore the effect of this kratom variety on pumping rate. As the pharyngeal muscle is similar to the mammalian heart muscle with a rhythmic pumping rate [60] any deviations in pumping parameters are a sensitive indication of toxicity [57]. Exposure to morphine significantly increased pumps per minute and reduced the interpump interval in both well-fed (Table 3 and Figure 6A) and starved worms (Table 4 and Figure 6B and Figure 7). White Borneo extracts at a concentration of 300 µg/mL did not alter the pump rate or interpump interval compared to the control, but the water extract did significantly alter pumping in starved worms (Table 4). The pump tracing of worms exposed to morphine was different from that of control and kratom-exposed worms (Figure 7), with the methanol extract causing the largest amplitude in contractions with the lowest frequency. It should be noted that DMSO had an effect in worms that were starved (Table 4 and Figure 6B). Here, the duration of pumps was significantly reduced, an effect not observed in well-fed DMSO-exposed worms.

## 4. Discussion

The research presented here focused on two specific areas of kratom pharmacology and toxicity that are not well-studied: (1) the toxicity of kratom extract and its constituents mitragynine and 7-hydroxymitragynine on reproduction and health of the progeny and (2) the evaluation of opioid-like effects of kratom compared to classical opioids such as morphine.

The concentration range used to evaluate each kratom extract was based on allometric scaling with 45 µg/mL in worms being approximately equivalent to a 5 g kratom dose in humans. The preclinical research of kratom is primarily conducted using ethanolic or methanolic extracts, whereas kratom is almost exclusively used by humans in water suspensions prepared as a tea for oral ingestion. For this reason, we chose to explore all three types of extraction in parallel. Our analysis of commercial kratom products indicates that both mitragynine and 7-hydroxymitragynine levels are higher in ethanolic and methanolic extracts compared to water extracts, which may in general explain the lower toxicity of kratom extracted in water. However, given the complex composition of kratom extracts with over 40 indole and oxindole alkaloids present, exploring the concentration of mitragynine and 7-hydroxymitragynine alone may not be sufficient to explain the pharmacology and toxicity of the extract. This study did indicate a higher toxicity of 7-hydroxymitragynine, which may be related to its higher potency at opioid receptors compared to mitragynine and morphine [11]. In addition, the variable amounts of 7-hydroxymitragynine in the samples may be related to generation of 7-hydroxymitragynine during the harvesting and processing of fresh leaves into the final powdered product [61].

Exposure to kratom did not result in any significant toxic effects on the nematodes. While elevated doses did adversely impact the worms, this was only in the methanolic and ethanolic extracts. As human consumption is via a water-based preparation, these effects are not likely to be observed in humans. However, between kratom varieties, there was a striking variety in the effects observed, with the White Borneo kratom consistently having a more severe impact on the health parameters tested compared to Bali and Red Maeng Da kratom. The phytochemical investigation was solely focused on the alkaloids mitragynine and 7-hydroxymitragynine in this study, while other alkaloids have been shown to contribute to pharmacological effects [62,63] both through opioid and non-opioid pathways. Further exploration of the presence of such alkaloids may indicate a distinction between White Borneo and the other studied kratom preparations.

Locomotor activity is widely used to assess the effects of morphine on animals, with morphine shown to alter mobility of rodents [64,65], crayfish [66], and snails [67], as well as *C. elegans* [68]. We confirmed that 0.5 mM morphine reduced the thrash rate of worms (to 40 body bends per minute), while at the same time, the methanolic and water extracts of White Borneo kratom reduced nematode mobility. There was no significant impact on nematode mobility upon exposure to other kratom extracts. As the impact on body bending was minimal compared to the morphine control, this suggests that kratom is unlikely to impact the opioid-like signalling pathway in the same way as morphine.

Food intake in mammals is also affected by opioids, with the opioid agonist, morphine, increasing food intake [69]. Similarly, the feeding behaviour of *C. elegans* is regulated by opioid peptides with morphine stimulating feeding in starved wild-type animals [40]. Our aim of assessing pump rate in *C. elegans* was to test if kratom stimulated pumping; if so, then kratom could be said to act in an opioid-like manner. We confirmed that upon activation by morphine in well-fed worms, as there was an increase in pump rate and a decrease in interpump interval, whereas all White Borneo kratom extracts at a concentration of 300 µg/mL did not significantly impact either parameter. Similarly, morphine impacted pumping in the same way in starved worms, and only the water extraction of kratom appeared to significantly affect pumping in these starved conditions. DMSO did cause a slight reduction in the pump duration in starved worms. This may be an artefact of the use of DMSO as a solvent, where there are short-term impacts on pumping at concentrations of DMSO above 0.5% [70]. As we used DMSO at 0.6%, it is possible that some of the effects observed were due to the solvent and not kratom. It would be if interest to further dissect the impact of DMSO and kratom on pumping parameters in starved worms. While morphine impacted the pumping parameters in both well-fed and starved conditions, an effect was only observed in the starved kratom-exposed worms; this suggests that kratom acts via an alternative pathway to opioid signalling. As pharyngeal pumping is a meaningful surrogate marker for neuronal toxicity [57,58] as well as mammalian heart functionality [60], our data suggest that kratom is unlikely to result in significant adverse effects on heart function in humans.

## 5. Conclusions

Kratom is thought to have physiological effects similar to those of opioids; however, the potential toxic risks to humans posed by kratom remain relatively unknown. The reported results from this study add to the existing literature on kratom pharmacology and toxicity, and to our knowledge, this is the first such investigation using *C. elegans* as a model. Taken together, the data from *C. elegans* suggest that kratom is not toxic and is also unlikely to act via the opioid-signalling pathway. Due to the conservation of biological processes between nematodes and humans, this would suggest that toxic effects observed in the nematode could be related to humans. Further, as molecular signalling and pathways are conserved [42,43], we are able to hypothesise that if kratom is unlikely to act via the opioid-signalling pathway in nematodes, this can be extrapolated to the molecular mechanism of kratom action in humans. However, further detailed analysis using cell-based assays to confirm the lack of action of kratom via opioid signalling in humans would be required, but this is beyond the scope of this current work. Ultimately, experiments using *C. elegans* have provided a unique opportunity to explore the toxicity of kratom. Further, the data suggest that kratom acts via an alternative pathway to opioid signalling; however, the full dissection of this pathway is deserving of future exploration.

## Figures and Tables

**Figure 1 ijerph-19-06294-f001:**
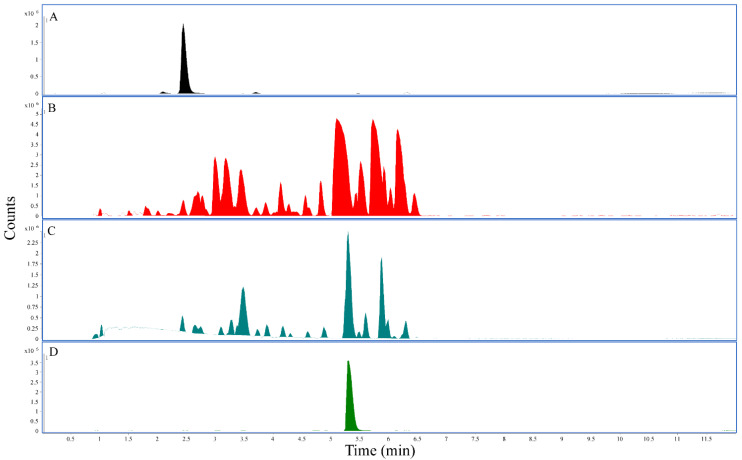
LC-QTOF chromatograms: All panels represent the base peak chromatogram. (**A**) 7-hydroxymitragynine standard, retention time 2.45 min. (**B**) The 25 mg/mL Red Maeng Da kratom methanolic extract. (**C**) The 25 mg/mL Red Maeng Da kratom aqueous extract. (**D**) Mitragynine standard, retention time 5.31 min.

**Figure 2 ijerph-19-06294-f002:**
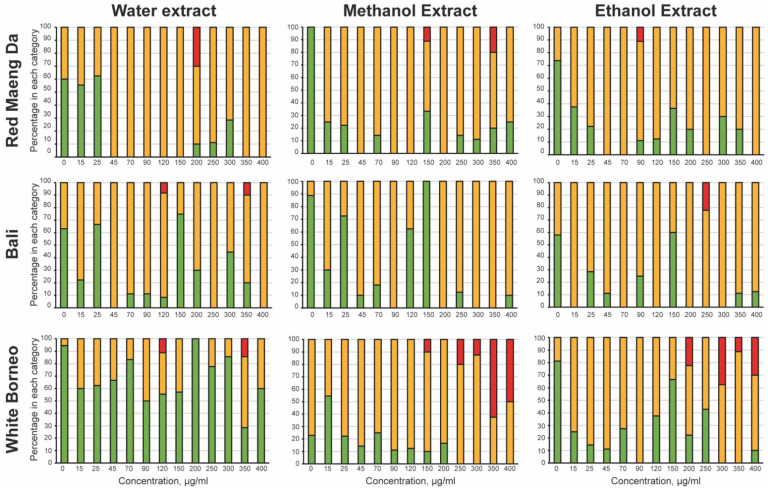
Acute toxicity of kratom towards the parental worms: The mobility of the parent was observed after 48 h of incubation with the kratom extracts. The worms were classified as ‘healthy’ (green bars), where the worm was freely moving with normal head-to-tail body bends or ‘slightly abnormal’ (yellow bars) where the worm had a defective thrashing motion. Any parental worms that were ‘dead/not moving’ (red bars) were also noted. The percentage of wells with the worms in each category was calculated and is shown as a stacked bar chart. White Borneo, Red Maeng Da, or Bali varieties extracted in water, methanol, or ethanol were added at 15, 25, 45, 70, 90, 120, 150, 200, 250, 300, 350, and 400 µg/mL.

**Figure 3 ijerph-19-06294-f003:**
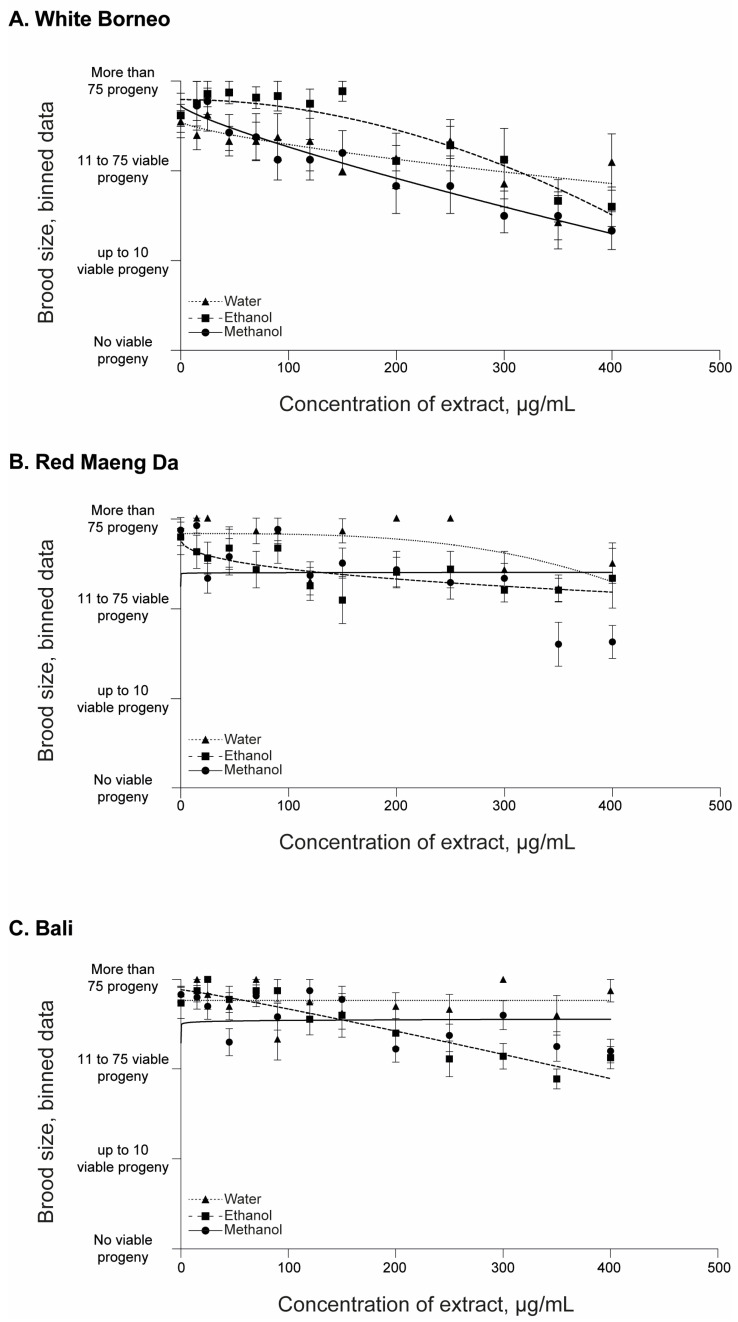
Brood size of *C. elegans* after 48 h of incubation with kratom: Worms were placed in liquid culture at the L4 stage and incubated at 20 °C for 48 h with shaking at 150 rpm. Kratom was added at 15, 25, 45, 70, 90, 120, 150, 200, 250, 300, 350, and 400 µg/mL with (**A**) White Borneo, (**B**) Red Maeng Da, or (**C**) Bali varieties extracted in ethanol (squares, dashed line), methanol (circles, solid line), or water (triangles, dotted line). The brood size was assessed by binning the data into 4 groups: no viable progeny; up to 10 viable progeny; 11–75 viable progeny; and >75 viable progeny (i.e., wild type). An average of the bins across the replicates was found and plotted with the standard error of the mean (s.e.m.) with the lines indicting the non-linear fit, with a hillslope of −1 using GraphPad Prism. Detailed information can be found in Appendix A.

**Figure 4 ijerph-19-06294-f004:**
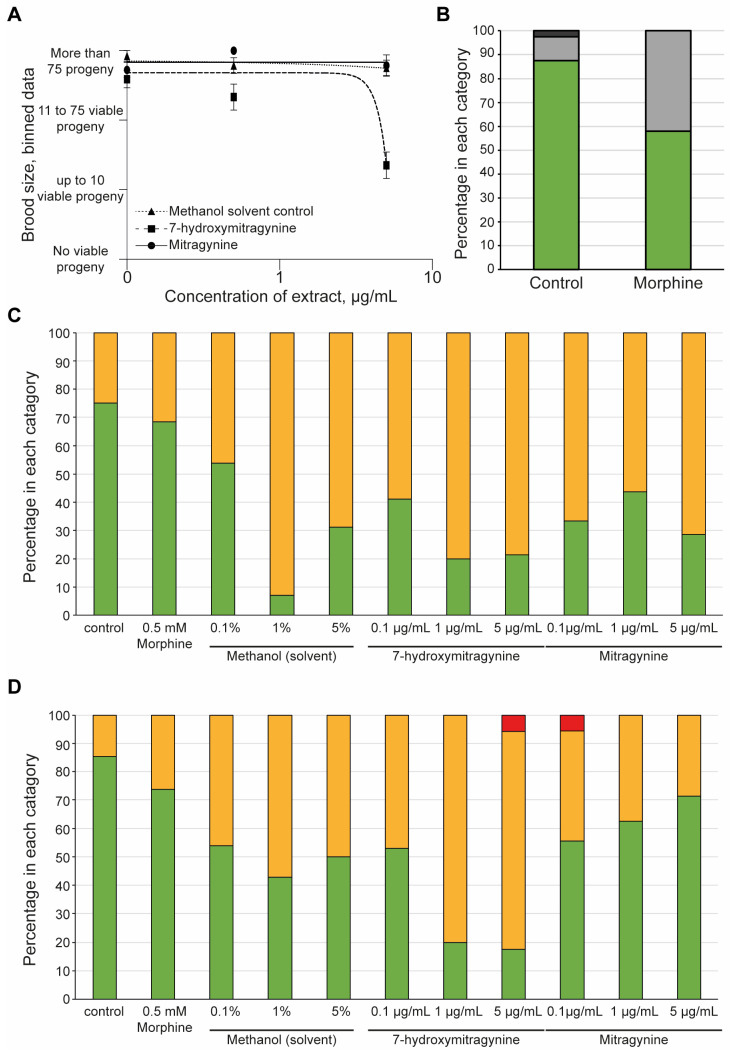
The effect of mitragynine, 7-hydroxymitragynine, and morphine on *C. elegans*: Both mitragynine and 7-hydroxymitragynine were tested at 5, 0.5, and 0.1 µg/mL in the assay, with the methanol control of 5%, 0.5%, and 0.1%, respectively. In this case, methanol was the solvent control as mitragynine and 7-hydroxymitragynine are not able to be fully dissolved in DMSO, and *C. elegans* can tolerate methanol up to 5% [53]. Morphine was only tested at 0.5 mM, as suggested in the literature [40]. (**A**) The brood size was assessed by binning the data into four groups: no viable progeny, up to 10 viable progeny, 11–75 viable progeny, or more than 75 viable progeny. An average of the bins across the replicates was found and plotted with the standard error of the mean and a non-linear regression line using GraphPad Prism. Mitragynine (circles, solid line) and the methanol control (triangles, dotted line) had no effect on the brood size of the worms. In contrast, 7-hydroxymitrgynine (squares, dashed line) resulted in a striking decrease in brood size at the highest concentration tested, 5 µg/mL. (**B**) Morphine did result in a decrease in brood, with around half of the replicates displaying a reduction in brood size. The brood size was assessed by binning the data into one of four classes: no viable progeny, red bars; up to 10 viable progeny (dark-grey bars); 11–75 viable progeny (light-grey bars); and more than 75 viable progeny (green bars). The percentage of wells in each category was calculated and plotted. (**C**) After 48 h of incubation with morphine, mitragynine, 7-hydroxymitrgynine, or methanol, the development of the progeny was assessed. The movement of the progeny was assessed in each well and classified as ‘healthy’ (green bars), ‘slightly abnormal (yellow bars), or ‘dead/not moving’ (red bars). The percentage of wells with offspring in each category at each concentration is shown. (**D**) The acute toxicity to the parent nematode was assessed after 48 h of incubation with morphine, mitragynine, 7-hydroxymitragynine, or methanol. The movement of the progeny was assessed in each well and classified as ‘healthy’ (green bars), ‘slightly abnormal’ (orange bars), or ‘dead/not moving’ (red bars). The percentage of wells with offspring in each category at each concentration is shown.

**Figure 5 ijerph-19-06294-f005:**
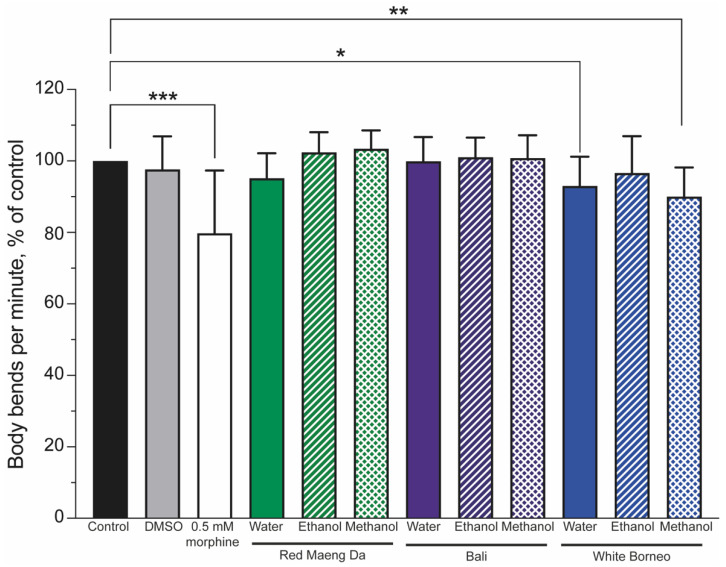
Thrashing was reduced in worms exposed to morphine and only the White Borneo variety of kratom: The thrashing of worms was normalised to control conditions (black bar), which was set to 100%. The vehicle control (0.6% DMSO, grey bar) showed no change in thrash rate, while 0.5 mM morphine (white bar) significantly reduced the body bends per minute in nematodes (*p* < 0.05, indicated by *, ** shows *p* < 0.01 and *** indicates *p* < 0.001). Neither Red Maeng Da kratom at 300 µg/mL (green bars) nor the Bali variety (purple bars) had a striking effect on body bends in worms with any of the extracts. There was a difference in the body bends observed when worms were exposed to the White Borneo kratom (blue bars). The water extract resulted in a small but significant decrease in body bends (*p* = 0.02, shown by *) with exposure to methanolic extracted White Borneo, resulting in a significant reduction in body bending (*p* = 0.002, shown by **). See Appendix A for individual counts of thrashing.

**Figure 6 ijerph-19-06294-f006:**
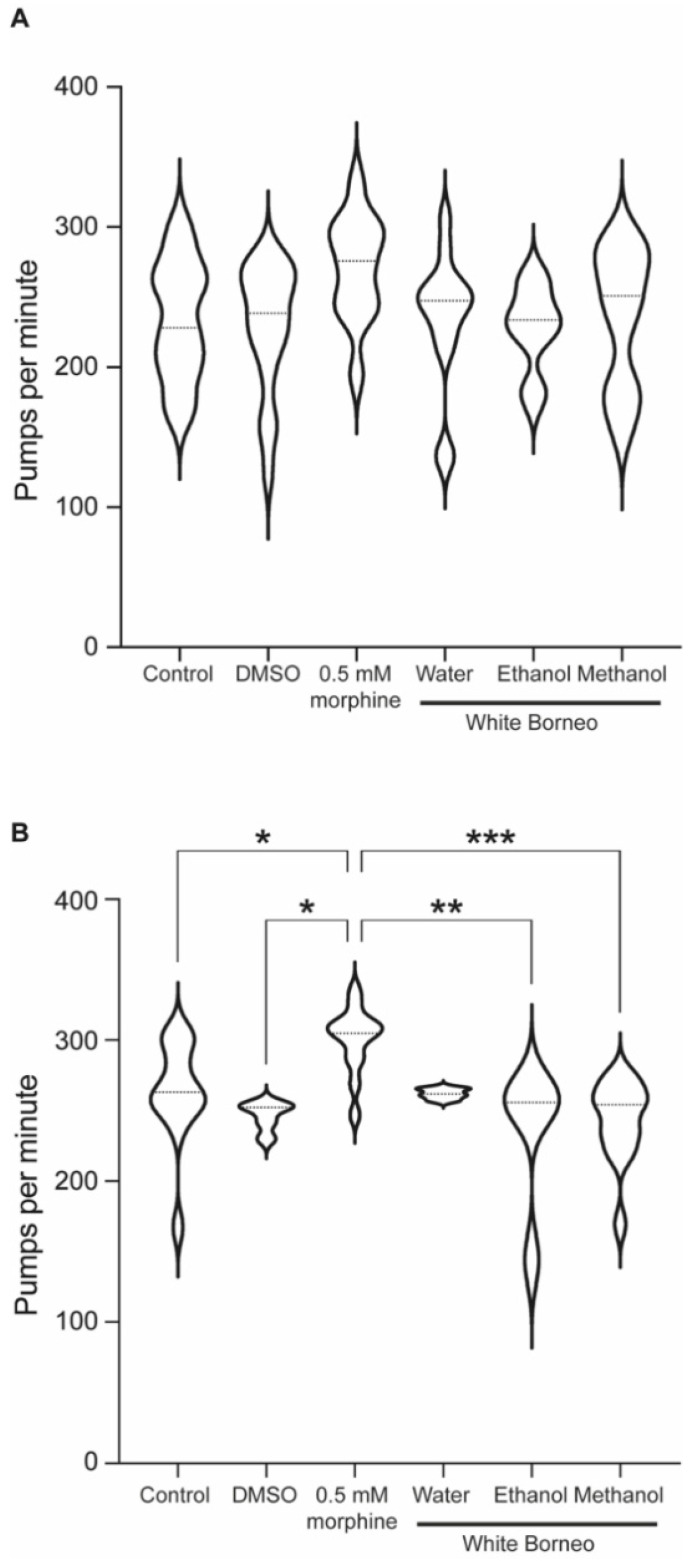
Morphine, but not kratom, increased the pumping rate in starved animals: Violin plot to show the pumps per minute for worms exposed to White Borneo kratom at 300 µg/mL, 0.5 mM morphine, or the controls. The violin plots also show the median value as a dotted line and statistics as shown from the one-way ANOVA test. (**A**) Well-fed worms. There are no statistical differences in these conditions. (**B**) Worms which were starved for 1 h. *p*-values are shown as a result of one-way ANOVA analysis where * *p* < 0.05, ** *p* < 0.005, and *** *p* < 0.0005.

**Figure 7 ijerph-19-06294-f007:**
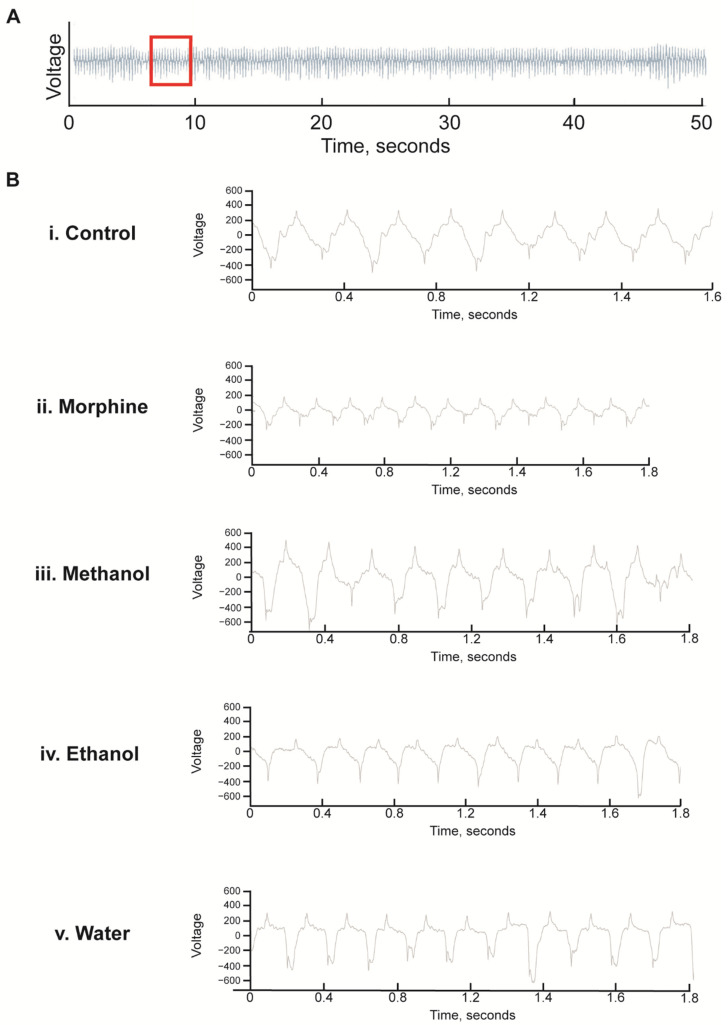
Representative electropharyngeogram traces of *C. elegans* exposed to White Borneo kratom with 1 h of starvation: Worms (L4 stage) were added to NGM supplemented with methanol, ethanol, or water extracts of White Borneo kratom (300 µg/mL) or 0.5 mM morphine and incubated at 20 °C. After 48 h, worms were transferred to kratom/morphine-supplemented NGM without food for 1 h before being subjected to analysis in the In Vivo Biosystem ScreenChip ™ system. (**A**) Representative trace of a control nematode for 50 s of a 2 min recording. The area of the red box, approximately 2 s, is expanded and shown in B. (**B**) Expanded traces of approximately 2 s showing the individual pumps. (**i**) Control nematodes; (**ii**) nematodes exposed to 0.5 mM morphine had more pumps which were smaller and shorter; (**iii**) Worms exposed to 300 µg/mL methanol extract of White Borneo kratom had pumps which were similar to the control; (**iv**) ethanol extract of White Borneo kratom at 300 µg/mL, and (**v**) White Borneo extracted in water had similar pumping profiles as the control animals. Detailed information on the EPGs is displayed in Table 4.

**Table 1 ijerph-19-06294-t001:** Repeatability and intermediate precision for mitragynine and 7-hydroxymitragynine standards. Coefficient of variation (% CV) provided for concentrations ranging from 10 ng/mL to 100 µg/mL. ND means not determined.

Concentration	%CV Mitragynine		% CV 7-Hydroxymitragynine	
	Intraday	Interday	Intraday	Interday
100 μg/mL	5.66%	12.93%	3.32%	6.85%
10 μg/mL	2.07%	5.28%	3.49%	9.59%
1 μg/mL	6.61%	12.44%	5.20%	11.91%
100 ng/mL	15.69%	16.60%	5.29%	8.58%
10 ng/mL	ND	10.29%	ND	14.85%

**Table 2 ijerph-19-06294-t002:** Concentrations of mitragynine and 7-hydroxymitragynine in commercial kratom products: Each kratom powder was extracted with ethanol, water, or methanol. Mitragynine concentrations are in μg/mg product and 7-hydroxymitragynine concentrations in ng/mg product.

Sample	Solvent	Mitragynine, μg/mg (%)	7-Hydroxymitragynine, ng/mg (%)
White Borneo	ethanol	36.44 (3.6%)	1.90 (0.002%)
	water	8.14 (0.8%)	0.0 (0.000%)
	methanol	37.01 (3.7%)	12.05 (0.012%)
Red Maeng Da	ethanol	36.48 (3.6%)	4.70 (0.005%)
	water	4.71 (0.5%)	10.16 (0.010%)
	methanol	34.28 (3.4%)	52.15 (0.052%)
Bali	ethanol	20.53 (2.1%)	18.28 (0.018%)
	water	1.37 (0.1%)	14.32 (0.014%)
	methanol	27.71 (2.8%)	11.11 (0.011%)

**Table 3 ijerph-19-06294-t003:** Pumping rates in well-fed L4 + 48 h-old nematodes: Worms were exposed to White Borneo kratom at 300 µg/mL, 0.5 mM morphine, or the controls from L4 stage. The asterisks indicate the results from the 2-tailed 2-sample *t*-test, where * *p* < 0.05, comparing the results from morphine exposure to the negative control. No asterisk is indicative of no significant difference. There was no significant difference in the comparison of the solvent control (DMSO) to any of the kratom extracts.

	Number	Pumps Per Minute	Pump Duration (Milliseconds)	Interpump Interval (Milliseconds)
Negative control	21	229	109	273
Solvent control (0.6% DMSO)	17	229	105	276
Morphine 0.5 mM	12	271 *	103	226 *
White Borneo Water extract	14	235	112	270
White Borneo Ethanol Extract	14	228	115	268
White Borneo Methanol Extract	15	235	109	268

**Table 4 ijerph-19-06294-t004:** Pumping rates in starved L4 + 48 h-old nematodes: Worms were exposed to White Borneo kratom at 300 µg/mL, 0.5 mM morphine, or the controls from L4 stage for 48 h. At this time, worms were transferred to unseeded NGM supplemented with morphine, DMSO, or kratom and left for 1 h before analysis. The asterisks indicate the results from the 2-tailed 2-sample *t*-test, where * *p* < 0.05 and ** *p* < 0.005, comparing morphine to the negative control and the kratom extracts to the DMSO solvent control. No asterisk is indicative of no significant difference. Plots to show the pumping rate are shown in Figure 5, and representative electropharyngeograms are displayed in Figure 6.

	Number	Pumps Per Minute	Pump Duration (Milliseconds)	Interpump Interval (Milliseconds)
Negative control	13	263	263	233
Solvent control (0.6% DMSO)	5	246	125 *	244
Morphine 0.5 mM	13	301 **	98	201 *
White Borneo Water extract	5	262 *	114 *	230 *
White Borneo Ethanol Extract	5	235	119 *	269
White Borneo Methanol Extract	13	243	124	252

## Data Availability

The data generated in this study are included in the article. Raw data that support the findings of this study are available from the corresponding author upon reasonable request. Some data may not be made available because of privacy or ethical restrictions.

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
