# Peer review of "Acute, Sublethal, and Developmental Toxicity of Kratom (Mitragyna speciosa Korth.) Leaf Preparations on Caenorhabditis elegans as an Invertebrate Model for Human Exposure"

_ijerph, 2022, doi:10.3390/ijerph19106294_

Round 1
Reviewer 1 Report
Overall study design and execution were good
- Typo error in title ‘yoxicity’
- In the title, the authors highlighted subchronic toxicity but, in the study, there was no subchronic toxicity conducted. Please clarify.
- The limitations include needing a further explanation as to why C. elegans are a good model for human development in regards to the morphological aspects they measured (e.g. if a compound causes body bends in C. elegans does this correlate to humans.
- Microwave-assisted extraction (MAE) does not seem to match how users would prepare their kratom.
- In the results section (page 6, line 239), the author mentioned methanol extracts of white Borneo and Bali have the highest mitragynine content, but Table 2 showed RMD has 34.28 µg/mg of mitragynine which is higher than Bali extract. Please clarify.
- Please include LC-QTOF chromatograms of White Borneo and Bali extract as well.
- In the results section (page 6, line 254), the author did not explain the result of Bali and RMD extracts.
- In Figure 3 and Figure 4A, Please include graph symbol legends beside the graph
- In the results section (page 11, line 330), the author mentioned both methanol and water extracts of White Borneo lowered body bends significantly but, the corresponding figure 5 showed only for White Borneo methanol extract. Please clarify.
- Legend of figure 5, the authors mentioned that both methanol and water extract significantly decrease body bends (p< 0.05) which is not tally with the graph. The author also mentioned methanol extracts have a very significant reduction in body bending (p<0.05), but the significant level was similar to water extract.
- The author mentioned that the solvent methanol did also impact brood size, and in the study, methanolic extracts showed the most toxicity effects. It is possible the toxicity effects observed in methanol kratom extracts could be partially due to the solvent.
- Solvent control (DMSO) also did decrease the average pump duration in starved worms, but the possible reasons are missing.
- In the discussion section (page 18, line 406), the authors could expand on why the white Borneo kratom are more toxic than Bali or RMD kratom in their experiments
- The authors could expand on why the 7-hydroxy alkaloid is more toxic than mitragynine in their experiments
- Supplemental Figure 3 can combine with Figure 4.
- In the discussion section (page 18, line 412), the authors mentioned kratom extracts had no significant impact on nematode mobility which was contradictory to the data reported (Figure 5).
- The authors could expand on why methanolic kratom extract did reduce the pump duration in staved worms
- In conclusion, it would be better if the author can highlight the major findings of the study (toxicity) which is lacking. Line 427-429: This sentence is confusing and took a few reads to understand. It would be helpful for the authors to clarify this sentence.
- The language needs to be tightened and the authors would benefit from an outside review to help with fixing minor English language issues
Author Response
1. Typo error in title ‘yoxicity’ This was an error in the title and we are very sorry for this oversight. This has been corrected.
2. In the title, the authors highlighted subchronic toxicity but, in the study, there was no subchronic toxicity conducted. Please clarify. The effects that we observe in the nematodes are subchronic, but sublethal is a better description. Therefore, we have changed the title to now say “acute, sublethal, and developmental toxicity”
3. The limitations include needing a further explanation as to why C. elegans are a good model for human development in regards to the morphological aspects they measured (e.g. if a compound causes body bends in C. elegans does this correlate to humans. The similarity of the musculature of the worm to vertebrates means that approaches to assess the physical performance of the worm using methods analogous to those used in human performance assessment can be used to assess the impact of toxic substances. Mobility is a powerful indicator of health and so body bending in C. elegans is a suitable readout of the health of the worm and can be used to assess the impact of drugs (see Anderson et al 2004).
Therefore, we have included the following text and reference at lines 504-503: Mobility is a powerful indicator of health and can be used to assess the impact of drugs [56].
The pharyngeal muscle generates a myogenic rhythm similar to that of mammalian heart muscle (Trojanowski et al 2016). Therefore, assessing deviations from pumping rate as a consequence of chemical exposure is a powerful and sensitive indicator of toxicity (Wellenberg et al. 2021). We have clarified this point at lines 545-547.
lines 545-547: As the pharyngeal muscle is similar to the mammalian heart muscle with a rhythmic pumping rate [60] any deviations of pumping parameters are a sensitive indication of toxicity [57].
4. Microwave-assisted extraction (MAE) does not seem to match how users would prepare their kratom. We have added language to clarify the choice of using MAE.
Lines 181-185 now reads as follows: Microwave assisted extraction (MAE) was used as a way to exhaustively extract semi-volatile and non-volatile compounds due to the increased efficiency and throughput coupled with decreased extraction time associated with this method. All three solvents were extracted in the same manner to limit additional influence of extraction efficiency inherent to the three solvents.
5. In the results section (page 6, line 239), the author mentioned methanol extracts of white Borneo and Bali have the highest mitragynine content, but Table 2 showed RMD has 34.28 µg/mg of mitragynine which is higher than Bali extract. Please clarify. We thank for the reviewer for their critical observation here. Indeed, this section must be clarified, and we have done so.
Lines 340-346 now read as follows: methanolic and ethanolic extracts prepared from White Borneo and Bali kratom powders presented with the highest concentration of mitragynine (Table 2). The highest concentration of 7-hydroxymitragynine was found in the methanolic extract for Red Maeng Da (0.052%) whereas Bali and White Borneo presented much lower concentrations of 7-hydroxymitragynine in the methanolic extract (0.011% and 0.012%, respectively). The water extracts were low in both mitragynine (0.1-0.8%) and 7-hydroxymitragynine (0-0.014%) (Table 2).
6. Please include LC-QTOF chromatograms of White Borneo and Bali extract as well. Figure 1 is representative of the observed chromatography of the extracts. We believe a figure showing all extraction combinations with 11 panels would be difficult to read while not adding to the main points of the study. We have added the following text at lines 348-351.
Lines 348-351 now reads: The representative chromatograms using Red Maeng Da extractions (Figure 1) further exemplifies the increased complexity of the alcoholic extractions compared to the more traditional water extraction.
7. In the results section (page 6, line 254), the author did not explain the result of Bali and RMD extracts. The reviewer is correct that we omitted to provide a full description of the results. To this end, we have modified the text at lines 380-386.
Lines 380-386 now reads as: Water extracts of White Borneo kratom was the least toxic to the parent, while exposure to water extracts of Bali and Red Maeng Da kratom caused some impairment at concentrations above 45 µg/mL. Both ethanolic and methanolic extracts of Bali and Red Maeng Da led to impaired movement in concentrations above 25 µg/mL. The methanolic and ethanolic extracts of White Borneo kratom resulted in a more striking effect on parent health, with exposure to these extracts causing death in parent worms above a concentration of 200 µg/mL.
8. In Figure 3 and Figure 4A, Please include graph symbol legends beside the graph We can see how the addition of a legend would aid in the clarity of the figures. We have therefore added a legend to each of the graphs in figure 3 and figure 4A. The new figures have been added to the manuscript.
9. In the results section (page 11, line 330), the author mentioned both methanol and water extracts of White Borneo lowered body bends significantly but, the corresponding figure 5 showed only for White Borneo methanol extract. Please clarify. Figure 5 contains the data from all extracts and all varieties of kratom tested, as well as a solvent control and a morphine positive control. Having double checked the analysis, we omitted the statistics for the White Borneo water extract. Figure 5 has now been updated with the associated figure legend. We have also altered the text to more fully represent the figure (lines 504-510) as well as including an additional figure that can be found as supplemental figure 3.
Lines 504-510: To this end, worms were exposed to controls (a DMSO solvent control and 0.5 mM morphine) or 300 µg/mL kratom for 48 hours. The number of body bends was assessed and normalised to the controls, such that the controls were set to 100%. Morphine significantly (p<0.001) lowered body bends (thrashing) compared to control animals (Figure 5A). Both the Red Maeng Da and Bali kratom preparations did not alter thrashing significantly in any solvent preparation, whereas White Borneo water and methanol extracts significantly lowered body bends (Figure 5 and Supplemental Figure 3).
10. Legend of figure 5, the authors mentioned that both methanol and water extract significantly decrease body bends (p< 0.05) which is not tally with the graph. The author also mentioned methanol extracts have a very significant reduction in body bending (p<0.05), but the significant level was similar to water extract. The reviewer was correct in pointing out our error here. While the water and methanol extracts of the White Borneo kratom do reduce body bending, we only included the statistics for methanol. To this end, we have updated the figure, figure legend and associated text (lines 505-510) and more detail can be seen in the response to comment #9.
11. The author mentioned that the solvent methanol did also impact brood size, and in the study, methanolic extracts showed the most toxicity effects. It is possible the toxicity effects observed in methanol kratom extracts could be partially due to the solvent. While this would be the initial thought, there is actually minimal solvent in the sample. The extractions were made in methanol (or ethanol or water) and then ultimately dried. The powders were then resuspended in DMSO for use in the worm experiments. DMSO up to 3% is not toxic to the worms (shown in previous publications, Boyd et al 2018 Tox Appl Pharmacol; Hughes et al. 2022 Pharmaceuticals). Further, within this manuscript we show that methanol does not impact brood size (Figure 4A) and while there is some impact on mobility, this is not dissimilar to control animals. As there is a reduction in the brood size with increasing concentrations of White Borneo kratom (Figure 3A) which is not observed in the Bali and Red Maeng Da varieties, nor in the methanol or DMSO controls, we believe the effect is a consequence of the kratom and the compounds found within.
12. Solvent control (DMSO) also did decrease the average pump duration in starved worms, but the possible reasons are missing. The reviewer is correct in highlighting this effect. However, the reduction is only observed in the starved worms and not in the well-fed worms that have been exposed to DMSO. Percentages of DMSO up to 0.1% do not affect the pumping however at 1% or more, DMSO is able to inhibit pharyngeal pumping in wild type animals, but the effect is short term (Calahorro et al. Toxicology Reports 2021). We agree with the reviewer that this should be noted in the manuscript, and to this end have added text in the results section at lines 554-557 and also the discussion at lines 676-691.
lines 554-557: It should be noted that DMSO has an effect in worms that are starved (Table 4 and Figure 6B). Here, the duration of pumps is significantly reduced, an effect not observed in well-fed DMSO exposed worms.
Lines 676-681: DMSO did cause a slight reduction in the pump duration in starved worms. This may be an artefact of the use of DMSO as a solvent, where there are short-term impacts on pumping at concentrations of DMSO above 0.5% [71]. As we use DMSO at 0.6%, it is possible that some of the effects observed are due to the solvent and not kratom. It would be if interest to further dissect the impact of DMSO and kratom on pumping parameters in starved worms.
13. In the discussion section (page 18, line 406), the authors could expand on why the white Borneo kratom are more toxic than Bali or RMD kratom in their experiments. We agree that this does require further exploration and added the following statement:
Lines 635-639: The phytochemical investigation was solely focused on the alkaloids mitragynine and 7-hydroxymitragynine in this study while other alkaloids have been shown to contribute to pharmacological effects [63,64] both through opioid and non-opioid pathways. Further exploration of the presence of such alkaloids may indicate a distinction between White Borneo and the other studied kratom preparations.
14. The authors could expand on why the 7-hydroxy alkaloid is more toxic than mitragynine in their experiments. We added a sentence that relates current pharmacology of 7-hydroxymitragynine to its potential higher toxicity:
Lines 624-626: This study did indicate a higher toxicity of 7-hydroxymitragynine which may be related to its higher potency at opioid receptors compared to mitragynine and morphine [61].
15. Supplemental Figure 3 can combine with Figure 4. We have now combined figures 4 and supplemental figure 3. The new figure has been included in the manuscript and the figure legend updated accordingly. We thank the reviewer for this suggestion to aid the understanding of the data.
16. In the discussion section (page 18, line 412), the authors mentioned kratom extracts had no significant impact on nematode mobility which was contradictory to the data reported (Figure 5). We thank the reviewer for highlighting this inconsistency. We have addressed the text so that it better reflects our analysis of the data and further highlights why we believe that kratom does not act in the opioid-signalling pathway.
Lines 642-647: We confirmed that 0.5 mM morphine reduced the thrash rate of worms (to 40 body bends per minute), while at the same time, the methanolic and water extracts of White Borneo kratom reduced nematode mobility. There was no significant impact on nematode mobility when exposed to other kratom extracts. As the impact on body bending was minimal compared to the morphine control, this suggests that kratom is unlikely to impact the opioid-like signalling pathway in the same way as morphine.
17. The authors could expand on why methanolic kratom extract did reduce the pump duration in staved worms The purpose of the pumping experiment was to test if kratom stimulated pumping – if it did, it was likely to be acting as an opioid (Cheong et al. 2015). However, we found that pump rate actually decreased in the presence of kratom and while this could be related to the DMSO concentration (see answer to comment 12), we think it is unlikely. The reason for a reduction in pumping in worms exposed to White Borneo methanolic extract in the absence of food is complex, and dissecting the mechanism(s) behind this is beyond the scope of this manuscript. However, we have made clear that our aim in this experiment was to test if kratom acted as an opioid (lines 670-671) and expanded our conclusions to this point (lines 671-687).
Lines 670-671: Our aim for assessing pump rate in C. elegans was to test if kratom stimulated pumping, if it did then kratom would act in an opioid-like manner.
Lines 671-687: We confirmed that upon activation by morphine in well-fed worms, there was an increase in pump rate and a decrease of inter pump interval, whereas all White Borneo kratom extracts at a concentration of 300 µg/mL did not significantly impact either parameter. Similarly, morphine impacted pumping in the same way in starved worms, and only the water extraction of kratom appeared to significantly affect pumping in these starved conditions. DMSO did cause a slight reduction in the pump duration in starved worms. This may be an artefact of the use of DMSO as a solvent, where there are short-term impacts on pumping at concentrations of DMSO above 0.5% [71]. As we use DMSO at 0.6%, it is possible that some of the effects observed are due to the solvent and not kratom. It would be if interest to further dissect the impact of DMSO and kratom on pumping parameters in starved worms. However, as morphine impacts the pumping parameters in both well-fed and starved conditions, but an effect is only observed in the starved kratom exposed worms, this suggests that kratom acts via an alternative pathway to opioid signaling. As pharyngeal pumping is a meaningful surrogate marker for neuronal toxicity [57,58] as well as mammalian heart functionality [60], our data suggests that kratom is unlikely to result in significant adverse effects on heart function in humans.
18. In conclusion, it would be better if the author can highlight the major findings of the study (toxicity) which is lacking. Line 427-429: This sentence is confusing and took a few reads to understand. It would be helpful for the authors to clarify this sentence. On reflection, we can see how this can be confusing. We have addressed this section and also added references to support our conclusion. Taking into account comments from other reviewers, the section (lines 688-704) has been updated accordingly.
Lines 688-704: Kratom is thought to have physiological effects similar to opioids, however, the potential toxic risks to humans posed by kratom remains relatively unknown. The reported results from this study add to the existing literature on kratom pharmacology and toxicity and to our knowledge is the first such investigation using C. elegans as a model. Taken together, the data from C. elegans suggests that kratom is not toxic and is also unlikely to act via the opioid-signalling pathway. Due to the conservation of biological processes between nematodes and humans, this would suggest that toxic effects observed in the nematode can be related to humans. Further, as molecular signalling and pathways are conserved [42,43] we are able to hypothesise that if kratom is unlikely to act via the opioid-signalling pathway in nematodes, this can be extrapolated to the molecular mechanism of kratom action in humans. However, further detailed analysis using cell-based assays to confirm the lack of action of kratom via opioid signalling in humans would be required, but this is beyond the scope of this current work. Ultimately, experiments using C. elegans have provided a unique opportunity to explore the toxicity of kratom. Further, the data suggests that kratom acts via an alternative pathway to opioid signalling, however the full dissection of this pathway is deserving of future exploration.
19. The language needs to be tightened and the authors would benefit from an outside review to help with fixing minor English language issues We have re-read the manuscript and addressed the level of English throughout. The manuscript is now in UK English.
Reviewer 2 Report
The manuscript “Acute, subchronic, and developmental toxicity of Kratom (Mitragyna speciosa Korth.) leaf preparations on Caenorhabditis elegans as an invertebrate model for human exposure” fits the journal’s scope. The authors present the acute and subchronic toxicity induced by a well-known herbal product – kratom on Caenorhabditis elegans nematode, as well as its influence on the development and physiological parameters.. The aim and the design of the research are clearly presented and justified. The methods and the results are presented in sufficient detail (some experiments in too much detail). The experiments were performed against several positive controls (morphine, mitragynine and 7-hydroxymitragynine). The manuscript is well written, and the quality of presentation is high. Overall, the manuscript is clear and well presented. However, before publication, the authors should clarify two major points:
- Although the aim of the study is clear, the novelty should be highlighted.
- As a consequence of the first point, the conclusions should be re-write. Ex.: lines 427-429; 430-432 – please rephrase; Please take into account that the mechanism of the herbal product and its major constituents were extensively researched – please see Eastlack, S. C., Cornett, E. M., & Kaye, A. D. (2020). Kratom—Pharmacology, clinical implications, and outlook: a comprehensive review. Pain and therapy, 9(1), 55-69.
Other minor comments:
Please correct the English/typing/formatting errors throughout the manuscript (including the title!)
Lines 47-51 – please add the reference(s)
Lines 92-95 - please correct the phrase
Author Response
The manuscript “Acute, subchronic, and developmental toxicity of Kratom (Mitragyna speciosa Korth.) leaf preparations on Caenorhabditis elegans as an invertebrate model for human exposure” fits the journal’s scope. The authors present the acute and subchronic toxicity induced by a well-known herbal product – kratom on Caenorhabditis elegans nematode, as well as its influence on the development and physiological parameters.. The aim and the design of the research are clearly presented and justified. The methods and the results are presented in sufficient detail (some experiments in too much detail). The experiments were performed against several positive controls (morphine, mitragynine and 7-hydroxymitragynine). The manuscript is well written, and the quality of presentation is high. Overall, the manuscript is clear and well presented. However, before publication, the authors should clarify two major points:
1. Although the aim of the study is clear, the novelty should be highlighted. We have added the following statement to the conclusions:
Lines 609-692: The reported results from this study add to the existing literature on kratom pharmacology and toxicity and to our knowledge is the first such investigation using C. elegans as a model.
2. As a consequence of the first point, the conclusions should be re-write. Ex.: lines 427-429; 430-432 – please rephrase; Please take into account that the mechanism of the herbal product and its major constituents were extensively researched – please see Eastlack, S. C., Cornett, E. M., & Kaye, A. D. (2020). Kratom—Pharmacology, clinical implications, and outlook: a comprehensive review. Pain and therapy, 9(1), 55-69. We thank the reviewer to bringing this review to our attention. We have included it in the introduction (new reference 3 in line 41 and line 67). Further, we have addressed this in the conclusion (along with comments from other reviewers) so that the conclusion (lines 688-704)
Lines 688-704: Kratom is thought to have physiological effects similar to opioids, however, the potential toxic risks to humans posed by kratom remains relatively unknown. The reported results from this study add to the existing literature on kratom pharmacology and toxicity and to our knowledge is the first such investigation using C. elegans as a model. Taken together, the data from C. elegans suggests that kratom is not toxic and is also unlikely to act via the opioid-signalling pathway. Due to the conservation of biological processes between nematodes and humans, this would suggest that toxic effects observed in the nematode can be related to humans. Further, as molecular signalling and pathways are conserved [42,43] we are able to hypothesise that if kratom is unlikely to act via the opioid-signalling pathway in nematodes, this can be extrapolated to the molecular mechanism of kratom action in humans. However, further detailed analysis using cell-based assays to confirm the lack of action of kratom via opioid signalling in humans would be required, but this is beyond the scope of this current work. Ultimately, experiments using C. elegans have provided a unique opportunity to explore the toxicity of kratom. Further, the data suggests that kratom acts via an alternative pathway to opioid signalling, however the full dissection of this pathway is deserving of future exploration.
Other minor comments:
3. Please correct the English/typing/formatting errors throughout the manuscript (including the title!) We apologise for the error in the title and have amended this, as well as other typographical errors throughout the manuscript.
4. Lines 47-51 – please add the reference(s) we have added additional references.
In line 58 we have added Kruegel et al. 2016 (new reference 6) and a review by Karunakarn et al 2022 (new reference 8).
In lines 59-60 we had added the following additional references: Kruegel et al. 2016; Takayama et al. 2002; Zhou et al. 2021 (references 6, 9 and 10 respectively).
At lines 61-62 to support the text we have added the following references: Obeng et al 2021; Takayama et al. 2002; Matsumoto et al 2004 (references 11, 9 and 12 respectively).
5. Lines 92-95 - please correct the phrase. We have rephrased this section as well as included new references, so that lines 96-100 now read as: follows
Lines 96-100: Due to the conservation at the cellular level with humans [31,41] as well as having conserved biological processes [42,43], the physiological responses of C. elegans following exposure to compounds, such as kratom is a good reflection of the overall response expected in humans, and provide key insights into the molecular mechanisms of toxicity [44].
Reviewer 3 Report
After reviewing the manuscript entitled: Acute, Subchronic, and Developmental Toxicity of Kratom (Mitragyna speciosa Korth.) Leaf preparations on Caenorhabditis elegans as an Invertebrate Model for Human Exposure. I consider that this original article is suitable for this special issue and is well structured. However, I have minor comments about some parts:
In title: You mean toxicity instead of yoxicity?
In abstract: In Keywords avoid terms repeated in title and abstract.
The references are not according to the MDPI style.
The introduction is well explained but you should include the main aim of this study.
In method: You included three commercial kratom powder products, but you need add some details like batch number and other specifications.
The main results showed in tables and figures support the conclusion.
Congratulations, it is a great work.
Author Response
After reviewing the manuscript entitled: Acute, Subchronic, and Developmental Toxicity of Kratom (Mitragyna speciosa Korth.) Leaf preparations on Caenorhabditis elegans as an Invertebrate Model for Human Exposure. I consider that this original article is suitable for this special issue and is well structured. However, I have minor comments about some parts:
1. In title: You mean toxicity instead of yoxicity? We apologise for the oversight of the spelling in the title of the manuscript, and indeed have changed yoxicity to toxicity
In abstract: In Keywords avoid terms repeated in title and abstract. We agree that the selection of key words could be better. To this end, we have added new key words.
New key words: Opioid; Toxicity; pharyngeal pumping; body bending; reproduction.
3. The references are not according to the MDPI style. This is true and an oversight from us. The reference style has now been updated to the MDPI style.
4. The introduction is well explained but you should include the main aim of this study. We have added a sentence stating the importance of the investigation at the end of the introduction and in the conclusions:
Lines 108-168 To our knowledge this is the first study to investigate toxicity of kratom in C. elegans as a feasible model to reflect human exposure.
Lines 690-692: The reported results from this study add to the existing literature on kratom pharmacology and toxicity and to our knowledge is the first such investigation using C. elegans as a model.
5. In method: You included three commercial kratom powder products, but you need add some details like batch number and other specifications.
Kratom is currently marketed and regulated in the US as a dietary ingredient that is not subject to the same strict regulations. At the time of writing this manuscript, the industry is self-regulating and not all companies follow GMP. The kratom products were purchased from a local smoke shop in Phoenix, AZ, and do not contain information such as batch number.
6. The main results showed in tables and figures support the conclusion. We have gone through the manuscript and ensured that the description in the text fully matches that in the table and figures. In places we altered the text to ensure clarity.
Congratulations, it is a great work.
Reviewer 4 Report
The subject of the paper has some relevance in the field as it looks at the opioid-like effect of kratom compared to morphine as well as at its toxicity. The manuscript is well written, clear, the introduction provides an adequate background to the topic, and the experimental section has the relevant information.
Nevertheless, some points need clarifying. On the issue of toxicity, the authors conducted experiments using three different extracts, aqueous, ethanolic and methanolic. But it is not altogether clear why they used alcoholic extracts if the substance is normally ingested in an aqueous form. Besides, the authors completely overlook the role of the solvent in the overall toxicity effect. Methanol and, to a lesser extent, ethanol, are extremely toxic substances. In an extraction operation like the one used in these experiments, the extracts are never completely solvent free. This is probably the reason for the methanolic extract having a much higher toxicity than the others, water in particular. However, this aspect has been completely overlooked by the authors. Therefore, this reviewer recommends this issue be addressed in the discussion section and the reasons for studying the alcoholic extracts in the first place be explained, in particular the methanolic one.
Furthermore, the authors do not explain the origin of the morphine used (commercial, extracted, etc…) neither the rationale for the concentration used. Also, the concentration unit (mM) used is different from all the other concentrations given in the manuscript which makes comparisons difficult. Please add some more information on this issue.
Finally, some format issues:
Line 63: the reference for Abdulla is not correctly given, it should be “Abdulla et al., 2021”. As a matter of fact, all the references need to be checked format-wise as, in general, the commas before the year are missing.
Section 2.1. Morphine, DMSO, NGM and E. coli were used in the experiment, but they are not mentioned in this section. Please add.
Line 155 & 206: E.coli should be italicized.
Table 1: format issues in the 1st and 2nd rows, bold is missing on some words.
Line 281: Please correct 20 ºC.
Author Response
The subject of the paper has some relevance in the field as it looks at the opioid-like effect of kratom compared to morphine as well as at its toxicity. The manuscript is well written, clear, the introduction provides an adequate background to the topic, and the experimental section has the relevant information.
1. Nevertheless, some points need clarifying. On the issue of toxicity, the authors conducted experiments using three different extracts, aqueous, ethanolic and methanolic. But it is not altogether clear why they used alcoholic extracts if the substance is normally ingested in an aqueous form. Besides, the authors completely overlook the role of the solvent in the overall toxicity effect. Methanol and, to a lesser extent, ethanol, are extremely toxic substances. In an extraction operation like the one used in these experiments, the extracts are never completely solvent free. This is probably the reason for the methanolic extract having a much higher toxicity than the others, water in particular. However, this aspect has been completely overlooked by the authors. Therefore, this reviewer recommends this issue be addressed in the discussion section and the reasons for studying the alcoholic extracts in the first place be explained, in particular the methanolic one. While this would be the initial thought, there is actually minimal solvent in the sample. The extractions were made in methanol (or ethanol or water) and then ultimately freeze dried. The powders were then resuspended in DMSO for use in the worm experiments. DMSO up to 3% is not toxic to the worms (shown in previous publications, Boyd et al 2018 Tox Appl Pharmacol; Hughes et al. 2022 Pharmaceuticals). Further, within this manuscript we show that methanol does not impact brood size (Figure 4A) and while there is some impact on mobility, this is not dissimilar to control animals. As there is a reduction in the brood size with increasing concentrations of White Borneo kratom (Figure 3A) which is not observed in the Bali and Red Maeng Da varieties, nor in the methanol or DMSO controls, we believe the effect is a consequence of the kratom and the compounds found within
2. Furthermore, the authors do not explain the origin of the morphine used (commercial, extracted, etc…) neither the rationale for the concentration used. Also, the concentration unit (mM) used is different from all the other concentrations given in the manuscript which makes comparisons difficult. Please add some more information on this issue. We have added the information concerning the morphine solution to section 2.1 (lines 177-179), including the concentration in mg/ml and mM. We used the concentration of 0.5mM for the nematode assays based on the literature (Cheong et al. 2015) and have included this information in the materials and methods section (lines 295-296). As far as we are aware there is no additional information as to the effect of morphine on elegans beyond the use of 0.5mM for pumping assays as described by Cheong et al. We agree that it is difficult to make comparisons between mM and mg/ml, however the purpose of including morphine was to ensure we had a positive control for the opioid pathway. Therefore, we believe that the comparison in terms of the effect of morphine with kratom is more relevant.
Lines 177-179: Morphine was purchased at 10 mg/mL morphine hydrochloride (Kalceks) in a physiological buffer solution. This equated to a 23.6mM solution.
Lines 295-296: A positive control of 0.5 mM Morphine was used based on literature [40]
Finally, some format issues:
3. Line 63: the reference for Abdulla is not correctly given, it should be “Abdulla et al., 2021”. As a matter of fact, all the references need to be checked format-wise as, in general, the commas before the year are missing. We apologise for the references being in an incorrect format, and we have address this so that they are all now in the MDPI style. We have also corrected the Abdulla reference accordingly.
4. Section 2.1. Morphine, DMSO, NGM and coliwere used in the experiment, but they are not mentioned in this section. Please add. We have not included NGM and E. coli in section 2.1 as these are nematode specific and details are found in section 2.4. We have included details of DMSO, serotonin and morphine in section 2.1.
Section 2.1 lines 176-179: For the nematode experiments, Serotonin hydrochloride was purchased from Sigma Aldrich (#H9523) as was DMSO (#D1435). Morphine was purchased at 10 mg/mL morphine hydrochloride (Kalceks) in a physiological buffer solution. This equated to a 23.6mM solution.
5. Line 155 & 206: colishould be italicized. This has been addressed throughout the manuscript.
6. Table 1: format issues in the 1stand 2nd rows, bold is missing on some words. We apologise for this oversight and have addressed the bold formatting.
7. Line 281: Please correct 20 ºC. We have corrected this.
Reviewer 5 Report
This study investigated alkaloid profile (mitragynine and 7-hydroxymitragynine) in 3 commercial kratom powder products using 3 different extraction solvents. The authors tested toxicity of the extracts using a C. elegans model. The results indicated compared to morphine, kratom extracts, especially the aqueous extract did not show significant toxicity.
The manuscript is, in principle, suitable for publication, but it needs some improvements prior to that.
- Title: correct “Yoxicity”
- Lines 137 and 138: What does the symbol between 2 m/z values mean? Please correct
- Table 1: in footnote, please spell out “ND”.
- Line 155: “E. coli” needs to be italic. Do a global search and correct them all, please.
- Can you convert Table 2 into a bar chart?
- Results 3.1: can you explain in the text, why methanol and ethanol extracts had so different results in terms of 7-hydroxymitragynine concentrations? Particularly, Red Maeng Da methanol had a very high 7-hydroxymitragynine concentration compared to others, why?
- Lines 244 – 255: the authors concluded that all water extracts have low mitragynine and 7-hydroxymitragynine, but this is not true for Bali. It has 14.32 ng/mg 7-hydroxymitragynine and that is comparable to methanol/ethanol extracts. Why?
- Results 3.2: “Kratom dose-dependently reduces brood size and health of parents and progeny”. From what’s presented in the paper, I’m not convinced the effects are dose dependent. Please change.
- Figure 2: Can you change orange colour to yellow? It’s not very well separated from red.
- Figure 2: can you explain in the text, which one indicates toxicity, only ‘dead/not moving’ counts or both ‘dead/not moving’ and ‘slightly abnormal” should be considered? Either way, I could hardly see a dose-dependent manner.
- Line 260: why it’s “30 μg/mL”? Please correct.
- Lines 274 – 275: “a decrease in brood size was observed with increasing concentrations,…”. Again, from what’s shown in Figure 3, this is NOT true. Maybe extracts of White Borneo are, but RMD methanol extract, Bali methanol and water extracts are clearly not the case.
- Figure 281: please correct the degree Celsius symbol.
- Line 299: Figure 4A?
- Lines 302 and 303: “Mitragynine did not present with the same effect although there were observed parent 302 deaths at the lowest concentration of 0.1 μg/mL”. Could you explain why?
- Figure 4: why uses methanol as the solvent control? Did you not use DMSO as solvent for the assay?
- Line 306: If you were going to use abbreviations for mitragynine and 7-hydroxymitragynine, please do so from the beginning of the paper.
- Lines 328 and 333: please correct “Figure 5A” and “Figure 5B”.
- Lines 350 – 351: “…but the methanolic extract did decrease average pump duration in starved worms”. I believe you should have compared values between extracts to the solvent blank (which should have been the negative control), not the negative control value in the current manuscript. There is no difference between the methanol extract and the solvent control in average pump duration.
- Lines 388- 389: order of the 2 specific areas should be turned around.
- Conclusion: I am not convinced that data in this study could lead to the conclusion that “kratom is unlikely to act primarily via the opioid-signaling pathway”. As far as I can see, this study only focused on toxicity of the kratom extract, which was found to be lower than what’s observed from morphine. I could not see any evidence of kratom extract showing its bioactivity similarly or differently to morphine as no such experiment has been done in this study. Please re-write conclusion based on what’s included in the manuscript.
- Please correct format of all references. They should be cited as numbers in the text.
Author Response
This study investigated alkaloid profile (mitragynine and 7-hydroxymitragynine) in 3 commercial kratom powder products using 3 different extraction solvents. The authors tested toxicity of the extracts using a C. elegans model. The results indicated compared to morphine, kratom extracts, especially the aqueous extract did not show significant toxicity.
The manuscript is, in principle, suitable for publication, but it needs some improvements prior to that.
1. Title: correct “Yoxicity” We apologise for the oversight of the spelling in the title of the manuscript, and indeed have changed yoxicity to toxicity
2. Lines 137 and 138: What does the symbol between 2 m/z values mean? Please correct We added clarifying text to explain the mass transitions used to quantify the kratom alkaloids.
Lines 210-214 now reads: The transitions used to quantify mitragynine was the precursor ion of 399.2 m/z fragmented into the product ion of 174.1 m/z (collision induced dissociation = 30V). The transition used to quantify 7-hydroxymitragynine was the precursor ion of 415.2 m/z fragmented into the product ion of 190.1 m/z (collision induced dissociation = 30V).
3. Table 1: in footnote, please spell out “ND”. We have added that ND means not determined to the figure legend.
4. Line 155: “E. coli” needs to be italic. Do a global search and correct them all, please. Apologies for this oversight, the text has been checked and all instances corrected.
5. Can you convert Table 2 into a bar chart? We intentionally expressed the alkaloid concentrations in table format so that the C. elegans data would be the focus of the article in easily displayed figures.
6. Results 3.1: can you explain in the text, why methanol and ethanol extracts had so different results in terms of 7-hydroxymitragynine concentrations? Particularly, Red Maeng Da methanol had a very high 7-hydroxymitragynine concentration compared to others, why?
Water as a solvent is not very efficient in extracting a range of natural products, among them alkaloids. Because the indole alkaloids are relatively large in terms of molecular weight and their charge is comparably small, their extraction is much more efficient in protic but more lipophilic solvents, like ethanol and methanol. The results obtained in this study are aligned with prior literature (Goh et al. 2021 Molecules 26(12):3704).
7. Lines 244 – 255: the authors concluded that all water extracts have low mitragynine and 7-hydroxymitragynine, but this is not true for Bali. It has 14.32 ng/mg 7-hydroxymitragynine and that is comparable to methanol/ethanol extracts. Why?
The variability in 7-hydroxymitragynine amount reflects what is being reported by vendors of kratom products over the years. Recently there has been discussion that 7-hydrxoymitragynine is actually generated during the harvesting and processing of the leaves and hence does not occur in the fresh leaf itself. This is one potential explanation of the higher concentration of 7-hydrxoymitragynine in the Bali extracts. We have added the following statement to the article:
Lines 627-629: In addition, the variable amounts of 7-hydroxymitragynine in the samples may be related to generation of 7-hydroxymitragynine during the harvesting and processing of fresh leaves into the final powdered product [62].
8. Results 3.2: “Kratom dose-dependently reduces brood size and health of parents and progeny”. From what’s presented in the paper, I’m not convinced the effects are dose dependent. Please change. We respectfully disagree with the reviewer, as we believe there are a dose dependent effect of the kratom extracts, but this is most evident with the White Borneo variety. We have made this more clear in the text of section 3.2, which now more accurately reflects the data that is observed (see also the responses to comments 10 and 12).
Section 3.2 lines 381-387: Water extracts of White Borneo kratom was the least toxic to the parent, while exposure to water extracts of Bali and Red Maeng Da kratom caused some impairment at concentrations above 45 µg/mL. Both ethanolic and methanolic extracts of Bali and Red Maeng Da led to impaired movement in concentrations above 25 µg/mL. The methanolic and ethanolic extracts of White Borneo kratom resulted in a more striking effect on parent health, with exposure to these extracts causing death in parent worms above a concentration of 200 µg/mL.
Lines 408-414: All parent worms were able to produce progeny, and the number of offspring was assessed in each exposure condition (Figure 3 and Supplemental Figures 1 and 2). There was a prominent decrease in the number of viable progeny in worms that were exposed to White Borneo kratom extracts, with the effects observed from 45 µg/mL (Figure 3A). In contrast, a small decrease in brood size was observed with increasing concentrations of Red Maeng Da and Bali extracts and these effects were observed at the higher concentration of exposures, from 350 µg/ml (Figure 3B, C).
9. Figure 2: Can you change orange colour to yellow? It’s not very well separated from red. We agree that the separation of red and orange is a challenge. Therefore, we have changed orange to yellow, as suggested. Consequently, we have changed all figures and the figure legends in the main text and in the supplemental figures.
10. Figure 2: can you explain in the text, which one indicates toxicity, only ‘dead/not moving’ counts or both ‘dead/not moving’ and ‘slightly abnormal” should be considered? Either way, I could hardly see a dose-dependent manner. The data was collected in a high-throughput manner and is used to give an estimation of the dose response seen. We feel that now we have altered the colour scheme (green, yellow and red, as requested in comment 9) a trend of toxicity is more clearly visible. There are clearly more red bars, indicative of acute toxicity, in the White Borneo samples.
Worms which die when exposed to the extracts are visualised under the microscope as having a rod-like structure, and if the worm is close to mortality, it does not move apart from infrequent “twitching”. These are shown by red bars. The yellow bars are worms with impaired movement, which is observed as a different pattern to the usual head-to-tail thrashing motion when worms are in a liquid culture. We have clarified this in the materials and methods section at lines 271-277.
Lines 271-277: As this is a method of screening for toxicity, the observation of each well was given a broad classification. For acute toxicity, the parent worms were classified into three groups: “Healthy” where the worm displayed normal development and head-to-tail body bending; “slightly abnormal” indicating that there was a small impact on mobility; and lastly “Dead or not moving” where the worm was observed as being dead by having a rod-like appearance, or close to death where there was very infrequent motion and usually only of the head.
11. Line 260: why it’s “30 μg/mL”? Please correct. This has been addressed (line 384). The correct value should have been 25µg/mL.
12. Lines 274 – 275: “a decrease in brood size was observed with increasing concentrations,…”. Again, from what’s shown in Figure 3, this is NOT true. Maybe extracts of White Borneo are, but RMD methanol extract, Bali methanol and water extracts are clearly not the case. We respectfully disagree with the reviewer on this point. We feel that there is a small decrease in all cases, but we do agree that the reduction is most striking in the case of exposure to White Borneo. Therefore, we have altered lines 409-414 accordingly.
Lines 409-414: There was a prominent decrease in the number of viable progeny in worms that were exposed to White Borneo kratom extracts, with the effects observed from 45 µg/mL (Figure 3A). In contrast, a small decrease in brood size was observed with increasing concentrations of Red Maeng Da and Bali extracts and these effects were observed at the higher concentration of exposures, from 350 µg/ml (Figure 3B, C).
13. Figure 281: please correct the degree Celsius symbol. Corrected.
14. Line 299: Figure 4A? We agree that adding the A, B or C after Figure 4 would aid in the interpretation of the text and figure. We have added this to the text in section 3.3.
15. Lines 302 and 303: “Mitragynine did not present with the same effect although there were observed parent deaths at the lowest concentration of 0.1 μg/mL”. Could you explain why? It is true that there is death of the parent worm at lower concentrations of mitragynine that is not observed in the controls. This is 6% of the wells which equates to just 1 animal in the whole set of replicates (where n=18), which is similar to the 5 µg/ml 7-hydroxymitragynine (1 worm out of 17 tested, 6%). This could be ‘bad luck” where that single worm died, or it could be an effect of the compound. However, the trend of toxicity is such that the effect of the single worm in minimal. We have added a sentence to explain this in the text at lines 453-456.
Lines 453-456: It should be noted that although parent deaths were observed at the lowest concentration of 0.1 µg/mL mitragynine and the highest concentration of 7-hydroxymitragynine (Figure 4C) this is in fact due to the death of a single worm.
16. Figure 4: why uses methanol as the solvent control? Did you not use DMSO as solvent for the assay? Methanol was required in this case as 7-hydroxymitragynine and mitragynine cannot be dissolved in DMSO. This was unfortunate, as elevated levels of methanol can be toxic to C. elegans. Worms can tolerate up to 4% methanol, with a 6% methanol having a severe impact on C. elegans (see Katiki et al. 2011). To clarify this, we have included the information in the figure legend to figure 4.
Figure 4 legend Lines 474-481: In this case, methanol was the solvent control as mitragynine and 7-hydroxymitragynine are not able to be fully dissolved in DMSO and C. elegans can tolerate methanol up to 5% [53]
17. Line 306: If you were going to use abbreviations for mitragynine and 7-hydroxymitragynine, please do so from the beginning of the paper. To ensure that there is consistency throughout the paper, we have altered figure 4 and the associated figure legend so that it says mitragynine and 7-hydroxymitragynine in full.
18. Lines 328 and 333: please correct “Figure 5A” and “Figure 5B”. Apologies for the oversight. We have corrected the text so that it only refers to the single figure 5 included.
19. Lines 350 – 351: “…but the methanolic extract did decrease average pump duration in starved worms”. I believe you should have compared values between extracts to the solvent blank (which should have been the negative control), not the negative control value in the current manuscript. There is no difference between the methanol extract and the solvent control in average pump duration. Based on the critical reading from the reviewers we have re-analysed the data from the pumping experiment. For the statistics in figure 6, we compared all conditions using one way ANOVA in GraphPad prism. While the graphical representations of the pumping are sound, we felt we could add more descriptive statistics and so have done this in a new figure and updated the figure legend accordingly.
For the tables, we used a 2-tailed 2-sample t-test to compare values. Based on the suggestions from the reviewer, we have updated how we compare the conditions, so that control vs morphine and solvent control vs kratom. To this end, there is no change for the results shown in table 3, however we have updated the figure legend to more accurately reflect the analysis. In contrast, there were minor changes to the analysis of the starved worms. Therefore, table 4 and its figure legend has been updated.
20. Lines 388- 389: order of the 2 specific areas should be turned around. We agree with the reviwers and so have swapped the two areas of focus.
lines 610-613: The research presented here focused on two specific areas of kratom pharmacology and toxicity that are not well studied: 1) the toxicity of kratom extracts and its constituents mitragynine and 7-hydroxymitragynine on reproduction and health of the progeny and 2) the evaluation of opioid-like effects of kratom compared to classical opioids like morphine.
21. Conclusion: I am not convinced that data in this study could lead to the conclusion that “kratom is unlikely to act primarily via the opioid-signaling pathway”. As far as I can see, this study only focused on toxicity of the kratom extract, which was found to be lower than what’s observed from morphine. I could not see any evidence of kratom extract showing its bioactivity similarly or differently to morphine as no such experiment has been done in this study. Please re-write conclusion based on what’s included in the manuscript. We acknowledge that our statements in the conclusion could be less bold. To this end, we have re-written the conclusion.
Conclusion lines 691-706: Kratom is thought to have physiological effects similar to opioids, however, the potential toxic risks to humans posed by kratom remains relatively unknown. The reported results from this study add to the existing literature on kratom pharmacology and toxicity and to our knowledge is the first such investigation using C. elegans as a model. Taken together, the data from C. elegans suggests that kratom is not toxic and is also unlikely to act via the opioid-signalling pathway. Due to the conservation of biological processes between nematodes and humans, this would suggest that toxic effects observed in the nematode can be related to humans. Further, as molecular signalling and pathways are conserved [42,43] we are able to hypothesise that if kratom is unlikely to act via the opioid-signalling pathway in nematodes, this can be extrapolated to the molecular mechanism of kratom action in humans. However, further detailed analysis using cell-based assays to confirm the lack of action of kratom via opioid signalling in humans would be required, but this is beyond the scope of this current work. Ultimately, experiments using C. elegans have provided a unique opportunity to explore the toxicity of kratom. Further, the data suggests that kratom acts via an alternative pathway to opioid signalling, however the full dissection of this pathway is deserving of future exploration.
22. Please correct format of all references. They should be cited as numbers in the text. We have addressed all the references throughout the manuscript to the correct MDPI formatting.
Round 2
Reviewer 2 Report
The authors corrected the manuscript according to the previous report, and the revised version is suitable for publication.
Reviewer 5 Report
Authors have sufficiently addressed all comments/questions.